# Non-cascade random walks in solid-state high harmonic generation

Zitan Zuo[1,4], Yiwen Wang[1,4], Shengzhe Pan [1] ✉, Lulu Han[1], Yidan Xu[1], Dian Wu[1], Shicheng Jiang [1] ✉ & Jian Wu [1,2,3] ✉

Random walks—both classical and quantum—unlocked new possibilities in search algorithms and information processing. Although linear photonic systems, with flexible tunability and multiple degrees of freedom, have served as efficient carriers for random walks, they typically require cascaded implementations, presenting a potential limitation on realizing integrated photonic circuits. In this work, we demonstrate a non-cascade, high-dimensional random walk in the orbital angular momentum (OAM) space of light using solid-state high-harmonic spectroscopy. The crystal nonlinearity enables the simultaneous conversion of multiple photons into a series of harmonics with distinct colors and whose OAM distributions are determined by the symmetry of the crystal. This approach reveals the dynamics of photonic degrees of freedom in high-harmonic generation can be naturally framed as an ultrafast, high-dimensional random walk, paving the way for compact, highly stable photonic platforms tailored for solid-state information processing.

The random walk, a cornerstone of stochastic process modeling, traces its origin to 1827 when Robert Brown observed the erratic motion of pollen grains suspended in water[1]. Its mathematical formalism was later proposed by Karl Pearson[2] and subsequently solidified by Albert Einstein through molecular kinetic theory[3]. In its simplest form, a discrete one-dimensional random walk describes propagation via sequential random steps, where a walker's position evolves probabilistically—akin to coin tosses—ultimately converging to a Gaussian distribution after many iterations. This framework has found broad applications across disciplines, from solid-state physics[4] and astronomy[5] to polymer chemistry[6], biology[7], computer science[8], and finance[9].

The quantum walk, a quantum-mechanical counterpart, is distinguished by its reliance on quantum superposition and the dispersed distributions. Experimental realizations have been demonstrated in diverse platforms, including superconducting systems[10–12], nuclear magnetic resonance[13,14], cold atoms[15], optical lattices[16,17], free electrons[18,19], and linear photonic systems[20–32]. Among these, the photonic system with flexible tunability and rich degrees of freedom[33–35] has been developed into an ideal carrier for quantum computation and

precision measurements. However, most photonic random walks rely on cascaded interferometric networks (Fig. 1a), where the physical footprint scales rapidly with step number[36]. This undoubtedly gives rise to inherent challenges in integration and scalability, as accumulated phase noise and environmental perturbations degrade performance.

Here, we introduce a non-cascade, high-dimensional random walk in orbital angular momentum (OAM) space of photons, enabled by solid-state high-harmonic generation (HHG)—a nonlinear up-conversion process that coherently generates radiation with multiples of the incident laser frequency[37–39]. The simultaneously emitted harmonics exhibit structured spatial and polarization profiles, where the harmonic order serves as an intuitive walking step. Unlike conventional cascaded photonic walks, our approach leverages the frequency-resolved, simultaneous emission of HHG (Fig. 1b), where the ultrafast electron dynamics in solids intrinsically encode the amplitude and phase for all walking steps at the same time. By exploiting the macroscopic symmetry of the crystal, we unveil a walking mode based on solid-state HHG, distinct from conventional paradigms. The combination of high-dimensional OAM space and non-cascade operation

[1]State Key Laboratory of Precision Spectroscopy, East China Normal University, Shanghai, China. [2]Collaborative Innovation Center of Extreme Optics, Shanxi University, Taiyuan, Shanxi, China. [3]Chongqing Key Laboratory of Precision Optics, Chongqing Institute of East China Normal University, Chongqing, China. [4]These authors contributed equally: Zitan Zuo, Yiwen Wang. ✉e-mail: szpan@lps.ecnu.edu.cn; scjiang@lps.ecnu.edu.cn; jwu@phy.ecnu.edu.cn

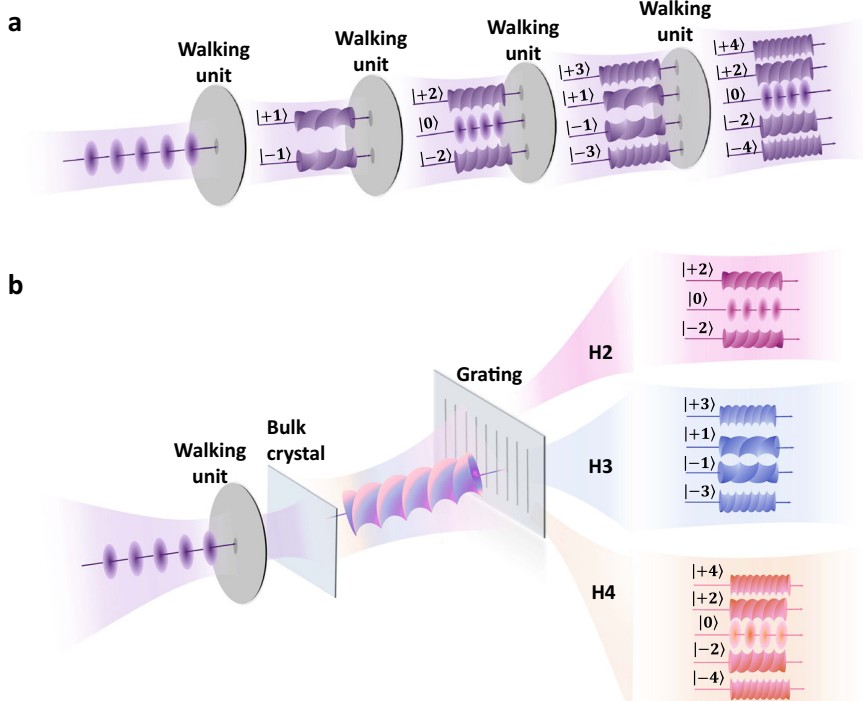

**Fig. 1 | Concepts for cascaded linear random walks and non-cascade high-harmonic random walks in photonic systems. a** Conventional cascaded implementation of quantum walks combines successive walking units, which enables random walks in the OAM space with the same laser frequency. **b** Non-cascade implementation of high-harmonic random walks in the OAM space. The initial fundamental wave for HHG is prepared by the same walking unit and focused into an inversion-symmetry broken crystal to generate both odd- and even-order harmonics. The frequency-resolved harmonics can be separated by a grating and the walking step is determined by the harmonic order. The OAM components of harmonic photons demonstrate the random walk in the infinite-dimensional OAM space.

of this approach establishes a route toward compact, ultrafast photonic systems for high-dimensional information processing.

## Results

### Construction of initial state for random walks

To implement the non-cascade random walks in photonic systems, we first identify two fundamental physical quantities of photons as vital subspaces: two-dimensional spin angular momentum (SAM) of $|s\rangle = |L\rangle$ or $|R\rangle$ as the coin space $\mathcal{H}_c$ and infinite-dimensional OAM of $|\ell\rangle = |-\infty\rangle \rightarrow |+\infty\rangle$ as the walker space $\mathcal{H}_w$ [22,23]. Compared to conventional cascaded random walks where a sequence of walking units $\hat{U}$ is requested to propagate the photonic system in the Hilbert space $\mathcal{H} = \mathcal{H}_c \otimes \mathcal{H}_w$ (Fig. 1a), simultaneous and individual different-order up-conversion processes with the absorption of multiple fundamental photons enable us to realize the non-cascade high-harmonic random walks (Fig. 1b). The absence of subsequent optical elements in non-cascade implementation necessitates pre-configuring the entire propagation operator in the input state. In this respect, the incident fundamental wave should carry the basic walking unit $\hat{U}$, including the conditional displacement of the walker and the coin-flip operation, which is implemented in experiments via a *q*-plate (QP) and a quarter-wave plate (QWP), respectively:

$$\hat{U} = \sum_{\ell=-\infty}^{\infty} \left[ \frac{1}{\sqrt{2}} (|R\rangle + i|L\rangle) \langle R| \otimes |\ell + 2q\rangle \langle \ell| + \frac{1}{\sqrt{2}} (i|R\rangle + |L\rangle) \langle L| \otimes |\ell - 2q\rangle \langle \ell| \right]. \quad (1)$$

Here, the topological charge $q = 1/2$ denotes the net displacement of $2q = 1$ in the OAM space after each step, $\ell$ denotes the OAM of the incident photon, and $L$ or $R$ denote the SAM of the incident photon. This walking unit points to that after each step of random walk, the walker still carries a new coin in hand and has a

uniform probability to decide the direction of the next step, which definitely constitutes the whole random walk process. Thus, the constructed fundamental wave is derived via the action of the walking unit on a linearly polarized Gaussian beam represented as $|\psi_0\rangle = (|L, 0\rangle + |R, 0\rangle)/\sqrt{2}$ (the notation $|s, \ell\rangle$ represents a direct product $|s\rangle \otimes |\ell\rangle$ of a certain SAM and OAM state for convenience), as shown in Fig. 1b. The experimental spatial profile of the fundamental wave with a central singularity (first row of Fig. 2a) is prepared in the spanned $2 \otimes 2$ Hilbert space as follows:

$$|\psi_{H1}\rangle = \hat{U}|\psi_0\rangle = \frac{i}{2}|L, +1\rangle + \frac{1}{2}|L, -1\rangle + \frac{1}{2}|R, +1\rangle + \frac{i}{2}|R, -1\rangle. \quad (2)$$

Despite the net OAM of the generated laser pulse being zero, it provides four eigenstates in both SAM and OAM subspaces with values of $|s| = |\ell| = 1$. Here, the relative phase among different angular momentum components is determined by the rotation angles of the QP and QWP. Although it introduces a global rotation in the far-field intensity profile (see Supplementary Note 1 for more details on the effect of relative phase), the probability distribution across OAM modes remains invariant, as it depends only on the square amplitude of each mode. These four fundamental photons with distinct SAMs and OAMs can be absorbed during the up-converted HHG process (see Methods for more details on the construction of incident vector beam). Although a coherent laser source rather than single-photon state is employed in experiments, the quantum characteristics of the random walk are preserved and fully captured through quantum interference among all possible pathways from a wave perspective [23,29,40-43].

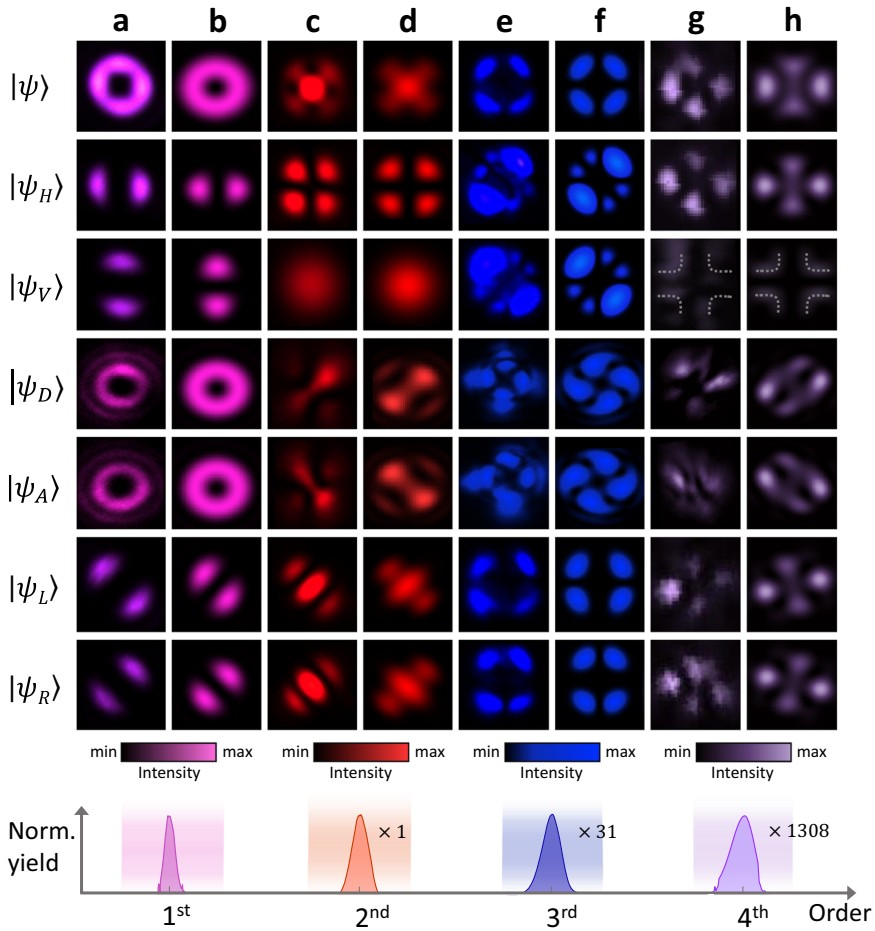

**Fig. 2 | Experimental observations and theoretical predictions of non-cascade high-harmonic random walks. a** Experimental spatial profiles of the fundamental wave with (first row) no polarization selection, (second row) horizontal component, (third row) vertical component, (fourth row) diagonal component, (fifth row) anti-diagonal component, (sixth row) left-handed component, and (seventh row) right-handed component. **b** Theoretically predicted spatial profiles of the fundamental wave. **c, e, g** Same as (**a**) but for the (**c**) second harmonic, (**e**) third harmonic, and (**g**) fourth harmonic, respectively. **d, f, h** Same as (**b**) but for the (**d**) second harmonic, (**f**) third harmonic, and (**h**) fourth harmonic predicted by the multi-photon absorption model, respectively. The vertical component of the fourth harmonic is an order of magnitude weaker than the other components and thus the white dashed curves are plotted to guide the eyes. The bottom inset is the normalized high-harmonic spectrum measured in experiments, and the normalization coefficients as the signature of conversion efficiencies are indicated on the right.

## Demonstration of non-cascade random walks

By focusing the fundamental wave $|\psi_{H1}\rangle$ into a bulk crystal of single-crystalline $\alpha$-quartz oriented along the $\langle 0001\rangle$ direction, we generate different-order harmonics as various steps of high-harmonic random walks. The inversion-symmetry-broken property of $\alpha$-quartz enables one to observe both odd- and even-order harmonics[44–47]. The spatial and polarization profiles of different-order harmonics are crucial for characterizing the corresponding SAM and OAM states through the high-harmonic random walks. Here, polarization-resolved measurements are performed to identify six distinct polarization states of the harmonics. The spatial profiles of fundamental wave (H1), second harmonic (H2), third harmonic (H3), and fourth harmonic (H4) are illustrated in Figs. 2a, 2c, 2e, and 2g, respectively, where the first row displays the overall profile without any polarization selection, and the other six display the polarization-resolved spatial profiles of $|\psi_H\rangle$, $|\psi_V\rangle$, $|\psi_D\rangle$, $|\psi_A\rangle$, $|\psi_L\rangle$, and $|\psi_R\rangle$. In contrast to the pumping beam with a single singularity, various structured profiles are generated from distinct walking steps, which are determined by the intrinsic symmetry of the $\alpha$-quartz. Taking the H2 emission as an example, the spatial profile with a central spot and four nodes around signifies the interference of the $\ell = 0, \pm 2$ components from the second step of high-harmonic random walks.

In order to gain insight into the characteristics of non-cascade high-harmonic random walks, it is essential to understand the nonlinear HHG process in the photon scenario with the principle of conservation of energy, momentum, and parity[48–53]. The incident vector laser pulse acting as a symmetric initial state in $2 \otimes 2$ angular momentum space can be divided as four fundamental waves with the same frequency and individual angular momentum components ($j = 1 \rightarrow 4$). Then, the harmonic frequency is described as the net absorption of the photons from the driving fields: $\Omega_n = \sum_j m_j \omega_j$ with $m_j$ and $\omega_j$ the absorbed photon number and frequency of the $j$-th driving photon, respectively, and $n = \sum_j m_j$. The OAM of the harmonics is proved to be the summation of the OAM carried by each involved photon, which reads $\ell_n = \sum_j m_j \ell_j$[54–57]. However, the SAM of the harmonic is more complicated due to the spin restriction of the emitted harmonic photon. The selection rule for SAM of the harmonics considering the $R$-fold rotational symmetry of solids is expanded as $s_n = \sum_j s_j + kR = \pm 1$ with $k$ an arbitrary integer[58,59]. The selection rule derived theoretically from both classical tensor calculation[60] and quantum Floquet theory[61] shows that the extra angular momenta of photons are transferred to the electron wave packets with certain macroscopic symmetry via the coupling between the light field and materials. Thus, the yield of the $n$-order harmonic illumination of

random walk can be described as

$$I_{HHG}(\Omega_n) \propto \left| \sum_{\Sigma m_j = n} \left[ \sigma_n \prod_j \left( e^{i\varphi_j} e^{i\ell_j\phi} E_{s_j}^{m_j} \right) \right] \right|^2 = \left| \sum_n \left( \sigma_n e^{i\varphi_n} e^{i\ell_n\phi} E_L^{m_1+m_2} E_R^{m_3+m_4} \right) \right|^2, \quad (3)$$

where $\sigma_n = \sigma(n; m_1, m_2, m_3, m_4)$ is the ratio of the multiphoton absorption pathways introduced from the number of permutation, $\varphi_j$ is the phase factor of each incident photon, $\varphi_n$ is the overall phase factor of a certain multiphoton absorption pathway expressed as $\varphi_n = \sum_j m_j \varphi_j$, $\phi$ is the azimuthal angle, and $E_{s_j}^{m_j}$ is the electric field of the $j$-th photon component.

During the multiphoton absorption process, photons carrying angular momentum superposition state are absorbed indiscriminately by the electron in the crystal, which can be regarded as an analog to random walks in photonic systems (Detailed derivation and discussion about random-walk behavior during successive photon absorption are elaborated in Supplementary Note 2). The SAM of each involved photon acts as a two-sided coin to decide the walking direction, and the OAM of the harmonic acts as a walker in the one-dimensional space. Compared to conventional random walks, the step number in our work is clearly represented by an additional dimension—the harmonic order. Here we take the fundamental wave, second harmonic, and third harmonic as examples, as shown in Fig. 3. The fundamental wave labeled H1 is the standard product of the step operator on the Gaussian mode at the zero point, resulting in four individual positions in the spanned $s \otimes \ell$ Hilbert space shown in Fig. 3c. The measured polarization-resolved spatial profiles shown in Fig. 2a are in great accordance with the theoretically simulated ones shown in Fig. 2b. The second harmonic labeled H2 illustrated in Fig. 3d demonstrates the two-photon up-converted process as a result of sequential evolution of two steps in the $s \otimes \ell$ space. Although there are nine possible pathways depicted in the Hilbert space, the pathways of absorbing both $s = \pm 1$ photons, namely, $0 \otimes \ell$, are forbidden according to the selection rule discussed above. Meanwhile, the SAM of the second harmonic that absorbs two fundamental photons with the same SAM is deduced to be opposite. Thus, the two-step second-harmonic emission is written as

$$|\psi_{H2}\rangle = (-|-2\rangle + 2i|0\rangle + |+2\rangle)|L\rangle + (|-2\rangle + 2i|0\rangle - |+2\rangle)|R\rangle. \quad (4)$$

We present the theoretically simulated spatial profiles in Fig. 2d as a direct comparison with the experimental measurements. In the OAM space, the walker moves from the zero point towards $(-1, +1)$ at the first step and $(-2, 0, +2)$ at the second step, and the walker at each OAM position has a two-sided coin at hand, which is actually the superimposed SAM states. This undoubtedly proves that the harmonics generated from a bulk crystal meet the fundamental criteria of the random walk and even expands the realization toward the frequency domain.

While for the third harmonic labeled H3 shown in Fig. 3e, the three-photon process is complex, and the populated Hilbert space becomes larger. Upon the absorption of three photons, there are sixteen possible pathways in the Hilbert space. However, the pathways of absorbing three identical SAM photons are forbidden considering the three-fold rotational symmetry of the $\alpha$-quartz. Therefore, the third-harmonic emission can be derived as

$$|\psi_{H3}\rangle = (i|-3\rangle - |-1\rangle + i|+1\rangle - |+3\rangle)|L\rangle + (-|-3\rangle + i|-1\rangle - |+1\rangle + i|+3\rangle)|R\rangle, \quad (5)$$

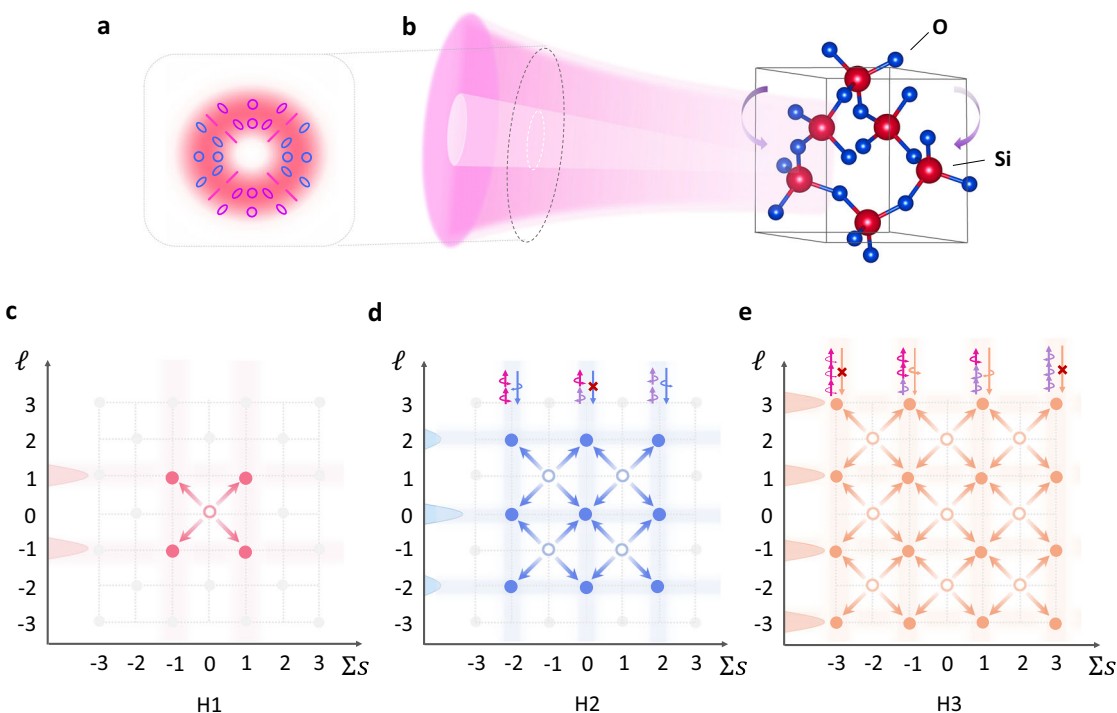

**Fig. 3 | Overview of non-cascade high-harmonic random walks in $\alpha$-quartz.** **a** Spatial and polarization profiles of the fundamental vector laser beam. The chirality of the elliptical (circular) polarization is colored with blue and purple. **b** Scheme of the interaction for HHG in three-fold rotational symmetric $\alpha$-quartz driven by a strong vector laser pulse. **c**–**e** Illustration of non-cascade random walks in the $s \otimes \ell$ Hilbert space for the (**c**) fundamental wave, (**d**) second harmonic, and (**e**) third harmonic, respectively. Hollow circles illustrate the positions of the previous step, while solid circles illustrate the positions of the present step. Upper insets denote the SAM selection rule of the harmonic emission via diverse multiphoton absorption pathways. Left insets denote the OAM population of the harmonic emission.

the simulated results from which are illustrated in Fig. 2f. The great agreement between the experiments and simulations proves that the walker randomly moves in the OAM space with a two-sided coin at hand, providing an extra knob to steer the high-harmonic random walks. The OAM space is expanded into $(-3, -1, +1, +3)$ after the three-step walk for the H3 emission. However, one can find that the walker still undergoes the complete second step during the H3 generation although it cannot reach the position of $0 \otimes \ell$ in H2 generation restricted by the selection rule. This, to our surprise, implies an advantageous non-cascade characteristic of the high-harmonic random walks in solids. All these random walks with different steps originate from the same initial state and will not interfere with the previous step. In contrast to the cascaded random walks in linear photonic systems, where sequential iterations are required, different-order high-harmonic random walks can be simultaneously generated.

### OAM distributions of high-harmonic random walks

Due to the non-cascade property of the high-harmonic random walks, an important feature of the implementation occurs that the distribution probability of this type of random walk shown in Fig. 4a is quite different from the ones of previous classical and quantum walks, acting as a potential choice of the walking algorithm. To validate the proposed high-harmonic random walks, we have performed experimental measurements to resolve the OAM components of individual harmonic signals based on spin-orbit tomography[62]. The measured OAM distributions for high harmonics in Fig. 4b shows a good agreement with the simulated results in Fig. 4a (see "Methods" and Supplementary Note 4 for more details on the spin-orbit tomography of high harmonics). One can observe a Gaussian distribution of classical walks in OAM space with the increment of walking step, as shown in Figs. 4c, d, while the distribution of quantum walks shows a clear

diffused characteristic in Fig. 4d. For the non-cascade random walks in solid-state HHG proposed here, the distribution demonstrates a complicated structure such as a flat-topped distribution for H3 in Fig. 4c and a diffused distribution for H4, the same as the four-step quantum walks, as shown in Fig. 4d. This can be understood as the synergy of the classical ratio stemming from multiphoton absorption pathways and the quantum selection rule depending on the macroscopic symmetry of the crystal.

Furthermore, our high-harmonic random walk exhibits a pronounced degree of programmability, achieved through active waveform manipulation of the driving laser field. For instance, by introducing a second-harmonic field with a specific polarization state, additional photon absorption pathways are activated, thereby modifying the output probability distribution. A theoretical demonstration of this programmability—using a two-color field to reconfigure the output states—is detailed in Supplementary Note 5. The adoption of an unbalanced coin with only a left-handed second-harmonic field leads to asymmetric walk distributions. Such control, extendable to other laser parameters like intensity and phase, underscores the potential of this non-cascade system for implementing diverse random walks and applications in parallel optical information processing.

## Discussion

In summary, we demonstrated non-cascade random walks in high-harmonic generation from crystalline solids. The frequency-resolved nature and the instantaneous absorption of multiple photons give birth to the distinct non-cascade characteristic of the high-harmonic random walks. Beyond its non-cascade property, the rotational symmetry of the crystal and the selection rule upon the SAM contribute to the rich spatial and polarization dependent distributions of the generated harmonics, by contrast with conventional cascaded random walks. The basic implementation in experiments only requests a QP, a

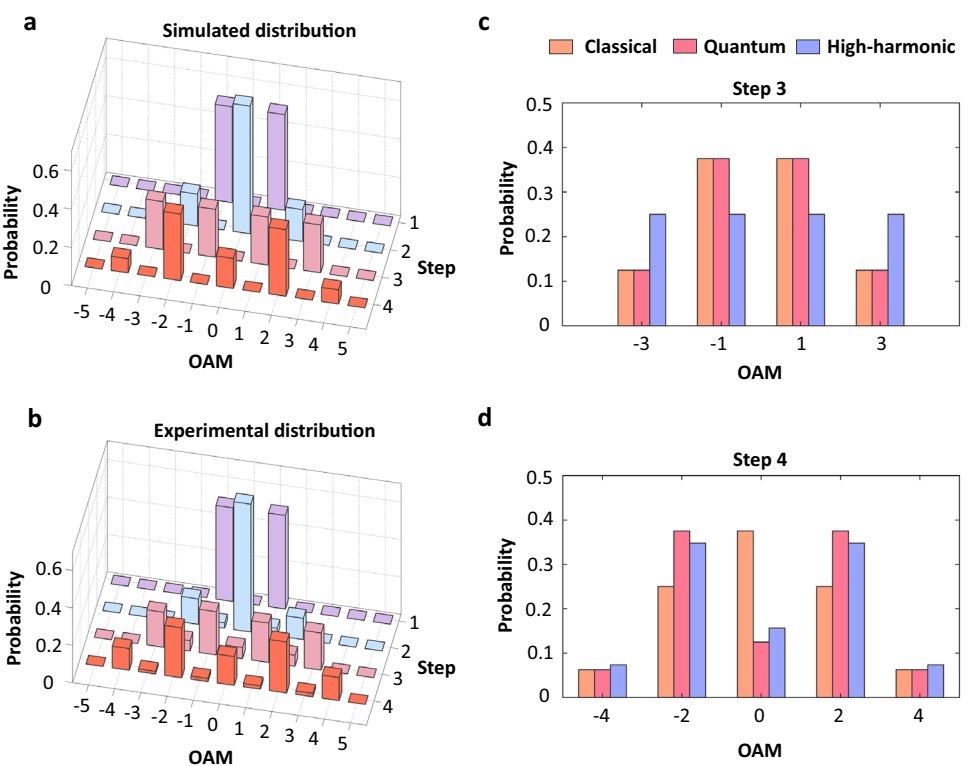

**Fig. 4 | High-harmonic random walks in OAM space. a** Simulated and **b** experimental non-cascade random walks generated by high-harmonic processes in a bulk crystal of α-quartz. **c** Flat-top distribution of the third-step high-harmonic

random walk (blue), compared to those from classical (orange) and quantum (pink) walks. **d** Diffused distribution of the fourth-step high-harmonic random walk (blue), compared to those from classical (orange) and quantum (pink) walks.

QWP, and a crystal with rotational symmetry, providing the potential to build a miniaturized, highly integrated and stable photonic system for non-cascade random walks. This architecture can be scaled to support a greater number of steps by leveraging established techniques—such as increasing the driving laser intensity and wavelength[37,39,44,63]. In this respect, we extend the concept of random walks toward the high-order harmonic generation in the frequency domain. The crystal with the inherent rotational symmetry is a natural random-walk integrator, working as the ideal platform for a unique type of random walk that is frequency-resolved, robust, and non-cascade. It sheds light on the solid-based walking simulation with the femtosecond iteration speed promised by HHG processes. Our work also presents an advanced demonstration of algorithmic simulations based on light-matter interactions, bridging the gulf between strong-field physics and quantum information and opening promising avenues for quantum computation in solid materials in the future.

## Methods
### Experimental technique
A linearly polarized infrared laser pulse centered at 1310 nm with a repetition rate of 10 kHz and a pulse duration of 45 fs (full width at half maximum) is produced from a commercial TOPAS-Prime in connection to a multipass amplified Ti-sapphire laser system. The linearly polarized Gaussian beam can be converted into a typical radially polarized beam with zero intensity at its center via passing through a QP with topological charge $q = 1/2$, which has been widely applied in the realm of quantum information and high-harmonic spectroscopy. The generated vector beam is a superposition of two standard Laguerre-Gaussian modes of $\ell = \pm 1$ with opposite circular polarizations of $s = \mp 1$. However, the radially polarized beam only has two components in the spanned $2 \otimes 2$ Hilbert space such as $|R, -1\rangle$ and $|L, +1\rangle$, which is asymmetric for the random walk in OAM space. To maintain the basic symmetry in the Hilbert space, we construct a complex vector beam using an additional QWP after the propagation through the QP. The QWP with $0°$ rotational angle acts as a Hadamard gate operator to perform a rotation operation in SAM subspace, giving birth to a superposition state of both right and left circular polarization. The spatial and polarization profiles of the generated vector beam are demonstrated in Fig. 3a. The polarization state evolves from a right-hand circular one to a linear one, and back to a left-handed circular one when the azimuthal angle rotates from 0° to 90° in the first quadrant. The polarization states of the other three quadrants hold mirror symmetry compared to the adjacent quadrants and a two-fold rotational symmetry in the whole polarization plane.

The spatially structured laser beam is then focused with 25-cm focus length on an 80 $\mu$m $\alpha$-quartz with the cut of $\langle 0001 \rangle$, resulting in a peak intensity of $\sim 5 \times 10^{13}$ W cm$^{-2}$ in the interaction region. The spatial structure of the crystal, belonging to the space group of P3$_1$21, is depicted in Fig. 3b. The bandgap of the $\alpha$-quartz is $\sim 9.5$ eV and the in-plane rotational symmetry is three-fold ($R = 3$). The inversion symmetry of the $\alpha$-quartz is broken and thus both odd- and even-harmonics are observed. The generated harmonics are propagated through bandpass filters and imaged by a CMOS camera. The polarization states of these harmonics are identified by a combination of a broadband QWP and a Glan prism.

To determine the OAM components of the generated harmonics, we perform modal decomposition using a spatial light modulator (SLM). This technique compensates for the OAM of the collimated harmonics by applying a helical phase factor $\exp(i\Delta\ell\phi)$, which transforms the initial state $(s, \ell)$ into a modified state $(s, \ell + \Delta\ell)$. The reflected harmonic beam is then Fourier-transformed by a lens and imaged onto a CMOS camera. The schematic of the experimental setup is shown in Supplementary Fig. 2. A distinct central spot is observed when the compensating topological charge $\Delta\ell$ equals the initial OAM value of the harmonic, enabling clear identification of the OAM component. For the

fourth harmonic, whose wavelength falls outside the operation range of the SLM, a series of spiral phase plates with different topological charges is used instead to provide the required wavefront modulation.

## Data availability
The data used in this study are available in the Zenodo database under accession code: https://doi.org/10.5281/zenodo.18396368.

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

## Acknowledgements

This work was supported by the National Natural Science Foundation of China (Grants Nos. 12521003, 12227807, 12241407, 12304378, 12504398, 12474487); the Science and Technology Commission of Shanghai Municipality (Grant Nos. 23JC1402000, 24YF2710000); the Shanghai Pilot Program for Basic Research (Grant No. TQ20240204).

## Author contributions

Z.Z., S.P., S.J., and J.W. conceived the idea of the research work. Z.Z., Y.W., S.P., L.H., Y.X., D.W., and J.W. performed the experiments. Z.Z., Y.W., S.P., and S.J. performed the theoretical study. All authors discussed the results and contributed to the writing of the manuscript.

## Competing interests

The authors declare no competing interests.
