## [Peer Review File · Nature Communications]

Non-cascade Random Walks in Solid-state High Harmonic Generation

Corresponding Author: Professor Jian Wu

Version 0:

Reviewer comments:

Reviewer #2

(Remarks to the Author)

The article describes high-harmonic generation in nonlinear crystals in terms of quantum walks. Specifically, it analyzes the spin and orbital angular momentum (SAM and OAM) spectrum of each harmonic and shows that, as the harmonic order increases, the corresponding state in the SAM–OAM space exhibits features characteristic of quantum walks when the step number is identified with the harmonic order. Experimental results are presented for output states containing up to four harmonic orders.

The system is introduced as a platform for realizing quantum walks in a non-cascaded configuration, since all step numbers are implemented within a single crystal rather than through a sequence of separate devices.

The general idea is interesting; however, in my opinion it does not meet the impact criteria required for publication in Nature Communications. To justify publication, the authors should provide stronger evidence that their approach constitutes a genuine alternative to standard cascaded platforms for optical information processing. Currently, this does not appear to be the case: the system does not seem scalable, it cannot be applied to single-photon states, and it lacks programmability. I will elaborate on these points in the detailed remarks below.

Major remarks

- In a quantum walk, the state at step j is obtained from the state at step $j-1$ through the action of a single-step unitary operator. In the present work, I see how the state at a given harmonic order is linked to the input state via n -photon absorption and selection rules. However, is this truly equivalent to a quantum walk? Is there a well-defined analogue of the single-step operator here, linking two states at consecutive time-steps?
- How programmable is this platform? In standard quantum-walk implementations, tuning the optical parameters of successive layers allows one to realize arbitrary walks. How could different types of quantum walks be accessed in the present scheme?
- What is represented on the horizontal axis of Figs. 3(C–E)? If s denotes the SAM index, it should only take two values. My understanding is that the walk occurs on a one-dimensional lattice spanned by the OAM modes.
- How scalable is this approach? What is the main factor limiting the harmonic order? How could higher orders be accessed?
- The spectra shown in Fig. 4 are only theoretical. An experimental measurement of these spectra would considerably strengthen the work, especially if the analysis is extended to different input states.
- Why are both a quarter-wave plate and a q-plate required to prepare an input state different from a single Gaussian beam with $l=0$?
- What is the role of the crystal symmetries? Would different symmetries lead to different walks?

For the reasons outlined above, I cannot recommend this manuscript for publication in Nature Communications. In my view, its main contribution lies in the interpretation of the high-harmonic generation spectrum within the framework of a quantum walk. After appropriate revisions, the work could be better published in a more specialized technical journal.

Reviewer #3

(Remarks to the Author)

The authors demonstrate a non-cascaded, high-dimensional random walk in the orbital angular momentum (OAM) space of light using solid-state high-harmonic spectroscopy. By utilizing a highly nonlinear process in a crystal, they achieve simultaneous multi-photon conversion into a series of harmonics at distinct wavelengths, with the OAM distribution governed by the crystal's symmetry. This approach provides a promising route toward compact and stable integrated photonic platforms for advanced information processing. As a fundamentally new strategy for realizing random walks without cascaded operations, this work exhibits high novelty and is likely to open new avenues for related research. Overall, the manuscript presents substantial value and will attract broad interest. The paper is also well-structured and supported by robust experimental data. I am pleased to recommend its publication in Nature Communications once the following major concerns are adequately addressed:

1. As is well known, the conversion efficiency typically decreases significantly with increasing harmonic order. Since the proposed non-cascaded random walk scheme relies on high-order nonlinear processes—where the harmonic order directly corresponds to the number of steps in the random walk—it is essential to include efficiency data in the main text. Providing either experimental or theoretical conversion efficiencies for each harmonic order would enable readers to better evaluate the practicality and scalability of this approach, and would facilitate future applications and follow-up studies by the community.
2. Although the authors have provided a description of the experimental methods in the latter part of the manuscript, the process remains somewhat challenging to follow—particularly given the multiple transformations involving orbital and spin angular momentum (OAM and SAM). To improve clarity and accessibility for a broader readership, it would be highly beneficial to include a schematic figure illustrating the experimental setup, with clear visual annotations of the state evolution throughout the process.
3. The manuscript does not provide a detailed explanation of the method used to measure the orbital angular momentum (OAM) intensity (or probability distribution) in the experimental setup. Given the central role of OAM in this study, it is essential to include a clear description of the measurement technique. Additionally, the potential influence of the relative phase among different OAM modes on the observed results should be briefly discussed. I guess the relative phase among different OAM modes will not influence the final results, since it is irrelevant to the probability?
4. A full characterization of polarization typically requires six measurements across the HV, $\pm 45^\circ$ (H \pm V), and circular (RL) bases. In Figure 2, however, intensity distributions are presented under only four polarization bases. Could the authors clarify whether a complete Stokes parameter measurement was deemed unnecessary in this context?

Version 1:

Reviewer comments:

Reviewer #2

(Remarks to the Author)

The authors have provided thorough and constructive responses to all comments raised by the referees. The manuscript has been significantly improved and now presents its main results in a clearer and more coherent manner. While I remain only partially convinced by the claims concerning the broader impact of the results, I find the core contribution solid and well supported.

I appreciate the novelty of the approach and the insight that the dynamics of photonic degrees of freedom in high harmonic generation can be naturally framed in terms of quantum walks. The experimental procedures appear sound and technically robust. On this basis, I recommend publication.

That said, I encourage the authors to revise the contextualization of their results. Beginning with the abstract, the comparison with the state of the art is presented from a somewhat misleading perspective. Although it is correct that cascaded systems face significant limitations for large scale integrated photonic circuits, I believe the present work should not be framed as a scalable alternative to such platforms. I am skeptical that step numbers exceeding those achievable with cascaded approaches can be realistically realized in this setting.

A clearer emphasis on the conceptual and physical innovation of the work, namely the emergence of a mechanism that can be interpreted as a quantum walk, together with the distinctive programmability enabled by pulse shaping techniques, would strengthen the manuscript and place its contributions in a more accurate context.

Reviewer #3

(Remarks to the Author)

All my concerns have been fully addressed, and the manuscript now meets the high standards required for publication in Nature Communications.

In terms of novelty, the authors leverage the intrinsic nonlinear optical response of a solid-state medium, using high-harmonic generation to simultaneously produce multiple harmonic orders within a single crystal. This unique approach enables parallel information processing in a compact, integrated, and inherently stable configuration, eliminating the need for cascaded components. The level of innovation demonstrated here is substantial enough to warrant publication.

Therefore, I strongly recommend that the manuscript be accepted for publication in Nature Communications.

Response to Reviewers' Comments:

In this response letter, we put the original *comments by the Reviewers in italics* to distinguish them from our responses in blue. Changes to the manuscript appear in red.

Reply to Reviewer #2

The article describes high-harmonic generation in nonlinear crystals in terms of quantum walks. Specifically, it analyzes the spin and orbital angular momentum (SAM and OAM) spectrum of each harmonic and shows that, as the harmonic order increases, the corresponding state in the SAM–OAM space exhibits features characteristic of quantum walks when the step number is identified with the harmonic order. Experimental results are presented for output states containing up to four harmonic orders.

The system is introduced as a platform for realizing quantum walks in a non-cascaded configuration, since all step numbers are implemented within a single crystal rather than through a sequence of separate devices.

The general idea is interesting; however, in my opinion it does not meet the impact criteria required for publication in Nature Communications. To justify publication, the authors should provide stronger evidence that their approach constitutes a genuine alternative to standard cascaded platforms for optical information processing. Currently, this does not appear to be the case: the system does not seem scalable, it cannot be applied to single-photon states, and it lacks programmability. I will elaborate on these points in the detailed remarks below.

Reply: We appreciate the Reviewer for recognizing the novelty of our general idea and for providing insightful comments regarding scalability and programmability in our non-cascade scheme. We agree that demonstrating a direct replacement for existing technologies is a high bar and would be a significant milestone. We would like to clarify, however, that the primary goal of this work is not to propose an immediate alternative for current devices, but to establish a new fundamental paradigm of non-cascade random walk via the multiphoton process of high-harmonic generation (HHG).

The approach proposed in our work is fundamentally distinct from standard cascaded quantum walks, which typically rely on linear optical elements arranged in cascaded interferometric networks. Such implementations face inherent challenges in scalability, as system expansion demands substantial physical space and highly stable environmental conditions. In contrast, our approach leverages the intrinsic nonlinear optical response of a solid-state medium, where HHG enables the simultaneous generation of multiple harmonic orders within a single crystal. This unique mechanism allows parallel information processing in a compact, integrated, and inherently stable configuration, without the need for cascaded components.

Our scheme circumvents several limitations of cascaded architectures and introduces a new paradigm for studying random-walk phenomena. This work helps bridge ultrafast science with information processing, and may inspire future developments in solid-state-based parallel computing—a point also appreciated by Reviewer 3.

We appreciate the Reviewer's comments regarding scalability, programmability, and applicability to single-photon states. While we will address scalability and programmability separately in subsequent responses, we would like to reply to the question regarding single-photon applicability. A single-photon source is not a necessary condition for constructing random walks. In optical implementations of random walks, the walking unit operates identically whether considered at the multiphoton level (i.e., coherent laser beams from a wave perspective) or at the single-photon level (i.e., single photons from a particle perspective). From the wave perspective, quantum characteristics in random walks are preserved through quantum interference among multiple possible pathways. This viewpoint is supported by a pioneering study by Goyal *et al.* [Phys. Rev. Lett. 110, 263602 (2013)], which demonstrated that random walks can be efficiently realized using coherent laser beams, interpreted via wave optics. Similar works have successfully employed coherent laser beams to implement random walks and related quantum-walk phenomena, including: Phys. Rev. A 75, 052310 (2007), Phys. Rev. Lett. 104, 050502 (2010), Phys. Rev. Lett. 106, 180403 (2011), Science 336, 55 (2012), Phys. Rev. Lett. 110, 263602 (2013), and Phys. Rev. Lett. 121, 100502 (2018). These studies confirm that the use of coherent light does not preclude the observation of essential quantum-walk features, as the underlying interference mechanisms are captured equally in a wave-based description.

To make it clear to the readers, we added the following sentences in the revised manuscript.

“Although a coherent laser source rather than single-photon state is employed in experiments, the quantum characteristics of the random walk are preserved and fully captured through quantum interference among all possible pathways from a wave perspective (23, 29, 40-43).” (Line 17 on page 4)

Comment #1: *In a quantum walk, the state at step j is obtained from the state at step $j-1$ through the action of a single-step unitary operator. In the present work, I see how the state at a given harmonic order is linked to the input state via n -photon absorption and selection rules. However, is this truly equivalent to a quantum walk? Is there a well-defined analogue of the single-step operator here, linking two states at consecutive time-steps?*

Reply: We thank the Reviewer for raising these important questions. We would like to clarify two key points in response.

First, as emphasized earlier, the high-harmonic random walk realized in our work—arising from nonlinear electron dynamics in a crystal—is not equivalent to a conventional quantum walk. Rather, it represents a distinct class of non-cascade random walk rooted in solid-state HHG process.

Second, although we don't have standard single-step operator linking different harmonic orders, there exists a well-defined equivalent that connects various photon absorption processes. Specifically, the HHG process can be interpreted in terms of cascaded N -photon absorption, governed by the following relation:

$$\beta^{(N)} = \left| \sum_{m_1+m_2+m_3+m_4=N} P_{m_1\sigma_1+m_2\sigma_2+m_3\sigma_3+m_4\sigma_4}^{(N)} \right|^2.$$

This can be recursively expressed as

$$P_{m_1\sigma_1+m_2\sigma_2+m_3\sigma_3+m_4\sigma_4}^{(N)} = C_1 P_{(m_1-1)\sigma_1+m_2\sigma_2+m_3\sigma_3+m_4\sigma_4}^{(N-1)} + C_2 P_{m_1\sigma_1+(m_2-1)\sigma_2+m_3\sigma_3+m_4\sigma_4}^{(N-1)} + C_3 P_{m_1\sigma_1+m_2\sigma_2+(m_3-1)\sigma_3+m_4\sigma_4}^{(N-1)} + C_4 P_{m_1\sigma_1+m_2\sigma_2+m_3\sigma_3+(m_4-1)\sigma_4}^{(N-1)}$$

where $P_{m_j\sigma_j}^{(N)}$ denotes the transition amplitude for absorbing m_j photons of angular momentum σ_j in generating the N -th harmonic, and C_j accounts for the number of quantum pathways leading to the transition. This formulation captures a cascaded absorption process, analogous to a step operator linking harmonic order $N-1$ to N . A similar interpretation has been established in the context of electron-light interactions, where sequential photon absorption gives rise to a random walk in the electron state space, as demonstrated in Science 373, eabj7128 (2021). In our solid-state system, the situation is further enriched: beyond multiphoton absorption, the process also involves high-harmonic photon emission. This enables simultaneous, non-cascade emission across multiple harmonic orders, distinguishing our platform from conventional step-by-step quantum walks while preserving an underlying recursive structure.

To make it clear to the readers, we rewrote the following sentence in the revised manuscript, and added a detailed discussion in Supplementary Materials.

“During the multiphoton absorption process, photons carrying angular momentum superposition state are absorbed indiscriminately by the electron in the crystal, which can be regarded as an analog to random walks in photonic systems (Detailed derivation and discussion about random-walk behavior during successive photon absorption are elaborated in Supplementary Materials).” (Line 8 on page 6)

Comment #2: *How programmable is this platform? In standard quantum-walk implementations, tuning the optical parameters of successive layers allows one to realize arbitrary walks. How could different types of quantum walks be accessed in the present scheme?*

Reply: We thank the Reviewer for this important point regarding programmability, which enables us to extend the approach. Indeed, our high-harmonic random walk platform can be effectively programmed using well-established optical-field-control techniques from ultrafast optics.

A representative example is introducing a second-harmonic (SH) laser field, which actively modulates the electron dynamics within the crystal and thereby enables access to different types of random walks. In addition to the pathways enabled by the fundamental laser field alone, the inclusion of an SH field with a state such as $|\psi_{\text{SH}}\rangle = |L, +2\rangle + |R, -2\rangle$ opens new photon absorption channels governed by

$$\Omega_n = \sum_j m_j \omega_j + \sum_k n_k (2\omega)_k.$$

Here, m_j and ω_j denote the number and frequency of absorbed j -th fundamental photon, n_k and $(2\omega)_k$ refer to the number and frequency of absorbed k -th SH photon.

To illustrate this programmability, we consider the generation of the third harmonic (H3) and assume equal contribution from two possible absorption pathways: $\omega + \omega + \omega$ and $\omega + 2\omega$. Under these conditions, the resulting H3 signal can be described as

$$|\psi_{H3}\rangle = (5i|-3\rangle - |-1\rangle + 3i|+1\rangle - 3|+3\rangle)|L\rangle \\ + (-3|-3\rangle + 3i|-1\rangle - |+1\rangle + 5i|+3\rangle)|R\rangle.$$

Figure R1a illustrates the resulting OAM distributions of H3 driven by a fundamental field (single-color in blue) and a fundamental field with an SH field (two-color in red), respectively. Notably, if the SH field contains only a left-handed circularly polarized component, i.e., effectively implementing an unbalanced coin, the output distribution becomes asymmetric, as yellow bars shown in Fig. R1a. Similarly, the corresponding OAM distributions of the fourth harmonic as the fourth-step random walk is shown in Fig. R1b.

This confirms that through appropriate tailoring of the driving laser fields—such as frequency, polarization, or intensity—our platform can be reconfigured to realize a range of distinct random-walk behaviors, thereby demonstrating a meaningful degree of programmability.

Fig. R1. Probability distribution of the (a) third-step and (b) fourth-step high-harmonic random walks driven by a single-color field (blue), a balanced two-color field (red), and an unbalanced two-color field (yellow), respectively.

Following the suggestion from the Reviewer and making it clear to the readers, we have added the following sentences in the revised manuscript, and added a detailed discussion in Supplementary Materials.

“Furthermore, our high-harmonic random walk exhibits a pronounced degree of programmability, achieved through active waveform manipulation of the driving laser field. For instance, by

introducing a second-harmonic field with a specific polarization state, additional photon absorption pathways are activated, thereby modifying the output probability distribution. A theoretical demonstration of this programmability—using a two-color field to reconfigure the output states—is detailed in Supplementary Materials. The adoption of an unbalanced coin with only a left-handed second-harmonic field leads to asymmetric walk distributions. Such control, extendable to other laser parameters like intensity and phase, underscores the potential of this non-cascade system for implementing diverse random walks and applications in parallel optical information processing.” (Line 19 on page 8)

Comment #3: *What is represented on the horizontal axis of Figs. 3(C–E)? If s denotes the SAM index, it should only take two values. My understanding is that the walk occurs on a one-dimensional lattice spanned by the OAM modes.*

Reply: We thank the Reviewer for the careful notice. The horizontal axis of Figs. 3(C–E) represents the accumulated spin angular momenta, denoted as $\sum S$, acquired through the absorption of multiple photons. To avoid any potential confusion, we have updated the axis label in the revised figures accordingly.

Comment #4: *How scalable is this approach? What is the main factor limiting the harmonic order? How could higher orders be accessed?*

Reply: We thank the Reviewer for raising this question regarding scalability. The scalability of our approach—specifically, access to higher harmonic orders—can be effectively achieved by increasing the intensity and wavelength of the driving laser field. The maximum attainable harmonic order is fundamentally limited by the relation $N_{max} \approx \Delta E / \hbar\omega$, where ΔE represents the energy difference between the valence band and the relevant conduction band to which an electron can be excited, and ω is the driving laser frequency. For instance, harmonic generation up to the 31st order has been demonstrated in MgO using such methods, as reported in *Opt. Lett.* **50**, 1492 (2025). The use of intense terahertz driving lasers could further extend this cutoff. This result strongly suggests that our non-cascade scheme can be directly extended to higher orders. With the ongoing advancement of intense mid-infrared femtosecond laser sources, accessing even higher-order harmonics is becoming increasingly feasible.

To clearly convey this point, we have added the following sentences in the revised manuscript.

“This architecture can be scaled to support a greater number of steps by leveraging established techniques—such as increasing the driving laser intensity and wavelength (37, 39, 44, 63)” (Line 11 on page 9)

Comment #5: *The spectra shown in Fig. 4 are only theoretical. An experimental measurement of these spectra would considerably strengthen the work, especially if the analysis is extended to different input states.*

Reply: We thank the Reviewer for the valuable suggestion. To further validate the proposed high-harmonic random walks, we have performed experimental measurements to resolve the OAM components of individual harmonic signals.

The OAM states of these harmonics are analyzed using a spatial light modulator (for the second and third harmonics) or custom-fabricated spiral phase plates (for the fourth harmonic) by applying a helical phase factor $\exp(i\Delta\ell\phi)$. This phase modulation transforms the collimated harmonic beam from an initial state (s, ℓ) into a modified state $(s, \ell + \Delta\ell)$. The reflected or transmitted beam is then Fourier-transformed by a lens and imaged onto a CMOS camera. A distinct central spot is observed when the compensating topological charge $\Delta\ell$ equals the initial OAM value of the harmonic, enabling clear identification of the OAM component.

Fig. R2. Experimental spin-orbit tomography for high-harmonic random walks. The top axis indicates the compensating topological charge $\Delta\ell$ applied via the spatial light modulator or spiral phase plates. Red squares highlight the configurations with distinct central spots, indicating the presence of a corresponding OAM state. Quantitative analysis of each OAM component is based on the integration of central zones in the images, marked by white dashed circles.

Figure R2 presents the experimental spin-orbit tomography for the high harmonic outputs. Although some discrepancies are observed, attributed to inhomogeneous harmonic emission patterns and the complexity of spin-orbit components, the measured OAM distributions show a good agreement with the simulated results, as summarized in Fig. R3.

Fig. R3. (a) Simulated and (b) experimental high-harmonic random walks.

Following the suggestion from the Reviewer, we added the following sentences in the revised manuscript, updated Figure 4, and added a detailed discussion in Supplementary Materials.

“To determine the OAM components of the generated harmonics, we perform modal decomposition using a spatial light modulator (SLM). This technique compensates for the OAM of the collimated harmonics by applying a helical phase factor $\exp(i\Delta\ell\phi)$, which transforms the initial state (s, ℓ) into a modified state $(s, \ell + \Delta\ell)$. The reflected harmonic beam is then Fourier-transformed by a lens and imaged onto a CMOS camera. A distinct central spot is observed when the compensating topological charge $\Delta\ell$ equals the initial OAM value of the harmonic, enabling clear identification of the OAM component. For the fourth harmonic, whose wavelength falls outside the operation range of the SLM, a series of spiral phase plates with different topological charges is used instead to provide the required wavefront modulation.” (Line 26 on page 10)

“To validate the proposed high-harmonic random walks, we have performed experimental measurements to resolve the OAM components of individual harmonic signals based on spin-orbit tomography (62). Figure 4b summarizes the measured OAM distributions for high harmonics, which shows a good agreement with the simulated results shown in Fig. 4a (see Methods and Supplementary Materials for more details on the spin-orbit tomography of high harmonics).” (Line 4 on page 8)

Comment #6: *Why are both a quarter-wave plate and a q-plate required to prepare an input state different from a single Gaussian beam with $l=0$?*

Reply: We thank the Reviewer for raising the question. In our non-cascade scheme, the entire multiple walk process is implemented within a single nonlinear crystal, rather than through a sequence of discrete optical elements. Consequently, the input state must be pre-configured to incorporate both the coin-flip operator in the SAM space (prepared by the quarter-wave plate) and the conditional shift operator in the OAM space (prepared by the q-plate). This ensures that spin and orbital states evolve simultaneously during the high-harmonic generation process. This approach of combining a quarter-wave plate and a q-plate to prepare the initial walker state is consistent with established protocols, as demonstrated, for example, in *Phys. Rev. Lett.* **110**, 263602 (2013).

To improve clarity in the revised manuscript, we have added the following sentence.

“The absence of subsequent optical elements in non-cascade implementation necessitates pre-configuring the entire propagation operator in the input state.” (Line 18 on page 3)

Comment #7: What is the role of the crystal symmetries? Would different symmetries lead to different walks?

Reply: We thank the Reviewer for raising the question. Indeed, crystal symmetry plays a fundamental role in our high-harmonic random walks, as it governs the selection rules that shape the underlying dynamics. In solid-state high-harmonic generation, the SAM selection rule follows $\sum_j s_j + kR = \pm 1$, where k is an integer and R denotes the rotational symmetry of the crystal. This rule, combined with OAM conservation, determines the accessible pathways and resulting walk behavior.

To illustrate this, we compare three representative wide-bandgap semiconductors with distinct rotational symmetries: MgO (<001>) with four-fold symmetry ($R=4$), ZnO (<0001>) with six-fold symmetry ($R=6$), and SiO₂ (<0001>) with three-fold symmetry ($R=3$). Using identical driving laser parameters as in the manuscript, we summarize in Table R1 the resulting OAM distributions of the third harmonic signal for each crystal. These results clearly show that different symmetries lead to distinct random walks—confirming that crystal symmetry serves as a key design parameter for tailoring random-walk behavior in our platform.

Table R1. OAM distributions for third harmonic using different crystals.

Crystal	Probability of OAM states			
	$ -3\rangle$	$ -1\rangle$	$ +1\rangle$	$ +3\rangle$
SiO ₂ (3-fold)	1	1	1	1
MgO (4-fold)	1	9	9	1
ZnO (6-fold)	0	0	0	0

For the reasons outlined above, I cannot recommend this manuscript for publication in Nature Communications. In my view, its main contribution lies in the interpretation of the high-harmonic generation spectrum within the framework of a quantum walk. After appropriate revisions, the work could be better published in a more specialized technical journal.

Reply: In summary, we sincerely thank the Reviewer for recognizing the novelty of our general idea and for providing these constructive suggestions, which have been invaluable in strengthening the presentation of our work and inspiring future applications.

Following the suggestions from the Reviewer, we have thoroughly revised the manuscript to:

- 1) Clarify the conceptual foundation of the non-cascaded scheme;
- 2) Demonstrate the scalability and programmability of the high-harmonic random walk platform;

3) Perform OAM-resolved measurements that confirm the predicted OAM distributions.

Fundamentally, this work extends—for the first time—the concept of a random walk into the high-harmonic generation regime, realizing a frequency-resolved, non-cascade walk within a single nonlinear crystal. It enables a robust form of walking that operates at the femtosecond timescales intrinsic to HHG. This approach establishes a new paradigm for ultrafast walking simulations in solids, beyond conventional cascaded architectures.

Furthermore, our results represent a novel demonstration of algorithmic simulation based on light-matter interaction, effectively bridging ultrafast physics with quantum information science. By leveraging the scalability and chip compatibility of solid-state systems—including bulk crystals, 2D materials, and metasurfaces—this platform opens promising pathways toward future on-chip quantum computation and information processing in the frequency domain.

We hope that the detailed responses and substantially improved manuscript adequately address the Reviewer's concerns, and we believe the work in its current form offers a timely and distinctive contribution suitable for publication in *Nature Communications*.

Reply to Reviewer #3

The authors demonstrate a non-cascaded, high-dimensional random walk in the orbital angular momentum (OAM) space of light using solid-state high-harmonic spectroscopy. By utilizing a highly nonlinear process in a crystal, they achieve simultaneous multi-photon conversion into a series of harmonics at distinct wavelengths, with the OAM distribution governed by the crystal's symmetry. This approach provides a promising route toward compact and stable integrated photonic platforms for advanced information processing. As a fundamentally new strategy for realizing random walks without cascaded operations, this work exhibits high novelty and is likely to open new avenues for related research. Overall, the manuscript presents substantial value and will attract broad interest. The paper is also well-structured and supported by robust experimental data. I am pleased to recommend its publication in Nature Communications once the following major concerns are adequately addressed:

Reply: We sincerely appreciate the Reviewer for the positive evaluation of our manuscript and for recommending its publication in *Nature Communications*. We are greatly encouraged by the Reviewer's assessment that our approach "provides a promising route toward compact and stable integrated photonic platforms for advanced information processing" and represents "a fundamentally new strategy for realizing random walks without cascaded operations."

In response to the Reviewer's valuable comments, we have provided detailed point-by-point responses below and have accordingly revised the manuscript to fully address all the suggestions raised.

Comment #1: *As is well known, the conversion efficiency typically decreases significantly with increasing harmonic order. Since the proposed non-cascaded random walk scheme relies on high-order nonlinear processes—where the harmonic order directly corresponds to the number of steps in the random walk—it is essential to include efficiency data in the main text. Providing either experimental or theoretical conversion efficiencies for each harmonic order would enable readers to better evaluate the practicality and scalability of this approach, and would facilitate future applications and follow-up studies by the community.*

Reply: We thank the Reviewer for this helpful suggestion. Following the suggestion, we have included the experimental conversion efficiencies for each harmonic order (prior to normalization) in the harmonic spectrum presented in Figure 2 and rewrote the following sentence in the revised manuscript.

"The bottom inset is the normalized high-harmonic spectrum measured in experiments, and the normalization coefficients as the signature of conversion efficiencies are indicated on the right." (Line 12 on page 20)

Comment #2: *Although the authors have provided a description of the experimental methods in the latter part of the manuscript, the process remains somewhat challenging to follow—particularly given the multiple transformations involving orbital and spin angular momentum (OAM and SAM). To improve clarity and accessibility for a broader readership, it would be highly beneficial to include a schematic figure illustrating the experimental setup, with clear visual annotations of the state evolution throughout the process.*

Reply: We thank the Reviewer for this helpful suggestion. Following the suggestion, we have added a schematic figure (shown here as Fig. R4) illustrating the experimental setup with clear visual annotations of state evolution in Supplementary Materials.

Fig. R4. Schematic of the experimental setup. An incident linearly polarized Gaussian beam is converted into a radially polarized beam by a q-plate (QP). The beam is then passed through a quarter-wave plate (QWP), resulting in a complex vector beam whose polarization ellipticity varies continuously with the azimuthal angle, cycling from linear to circular and back to linear. This vector beam is focused by a lens onto a bulk α -quartz crystal to generate high harmonics. Each harmonic beam can be selected using a band-pass filter. The polarization state of the selected harmonic is analyzed using a combination of a QWP and a polarizer. Meanwhile, its OAM components are characterized by diffraction from a spatial light modulator (SLM). The resulting beam profile is finally focused onto a CMOS camera for detection.

Comment #3: *The manuscript does not provide a detailed explanation of the method used to measure the orbital angular momentum (OAM) intensity (or probability distribution) in the experimental setup. Given the central role of OAM in this study, it is essential to include a clear description of the measurement technique.*

Reply: We thank the Reviewer for this helpful suggestion. To prove the reliability of our results, we have added the description of OAM measurement technique in the revised manuscript.

“To determine the OAM components of the generated harmonics, we perform modal decomposition using a spatial light modulator (SLM). This technique compensates for the OAM of the collimated harmonics by applying a helical phase factor $\exp(i\Delta\ell\phi)$, which transforms the initial state (s, ℓ) into a modified state $(s, \ell + \Delta\ell)$. The reflected harmonic beam is then Fourier-transformed by a lens and imaged onto a CMOS camera. A distinct central spot is observed when the compensating topological charge $\Delta\ell$ equals the initial OAM value of the harmonic, enabling clear identification of the OAM component. For the fourth harmonic, whose wavelength falls outside the operation range of the SLM, a series of spiral phase plates with different topological charges are used instead to provide the required wavefront modulation.” (Line 26 on page 10)

Comment #4: Additionally, the potential influence of the relative phase among different OAM modes on the observed results should be briefly discussed. I guess the relative phase among different OAM modes will not influence the final results, since it is irrelevant to the probability?

Reply: We thank the Reviewer for raising the question. In the multiphoton absorption process of HHG, the phase coherence of the fundamental driving laser is inherently preserved and directly transferred to the emitted harmonics. Taking the fundamental beam as an example, when the rotation angle of QP is denoted as θ and the rotation angle of QWP is denoted as β , the constructed fundamental beam can be expressed as

$$|\psi_{H1}\rangle = \frac{i}{2}e^{i\theta}|L, +1\rangle + \frac{1}{2}e^{-i(\theta+2\beta)}|L, -1\rangle + \frac{1}{2}e^{i(\theta-2\beta)}|R, +1\rangle + \frac{i}{2}e^{-i\theta}|R, -1\rangle.$$

The resulting second harmonic beam is given by

$$|\psi_{H2}\rangle = (-e^{-2i\theta}|-2\rangle + 2ie^{-2i\beta}|0\rangle + e^{i(2\theta-4\beta)}|+2\rangle)|L\rangle + (e^{-i(2\theta-4\beta)}|-2\rangle + 2ie^{2i\beta}|0\rangle - e^{2i\theta}|+2\rangle)|R\rangle.$$

As correctly noted by the Reviewer, although the relative phase among OAM modes influences the spatial structure of the harmonic field—resulting in a global rotation of the far-field intensity profile, as illustrated in Fig. R5—it does not alter the probability distribution across OAM modes. This is because the observable probability in the OAM basis depends only on the squared amplitude of each mode and remains invariant under global rotation or relative phase shifts. Therefore, while the relative phase can modulate the spatial arrangement of the output, it does not affect the core statistical outcomes of the random walk, which are the focus of this study.

Following the suggestions from the Reviewer, we have added the following sentence in the revised manuscript and added a detailed discussion in Supplementary Materials.

“Here, the relative phase among different angular momentum components is determined by the rotation angles of the QP and QWP. Although it introduces a global rotation in the far-field intensity profile (see Supplementary Materials for more details on the effect of relative phase), the probability distribution across OAM modes remains invariant, as it depends only on the square amplitude of each mode.” (Line 10 on page 4)

Fig. R5. Influence of relative phase among OAM modes on the resulting second-harmonic intensity profiles. White dashed lines indicate the corresponding global rotation of the far-field patterns.

Comment #5: A full characterization of polarization typically requires six measurements across the HV, $\pm 45^\circ$ ($H\pm V$), and circular (RL) bases. In Figure 2, however, intensity distributions are presented under only four polarization bases. Could the authors clarify whether a complete Stokes parameter measurement was deemed unnecessary in this context?

Reply: We thank the Reviewer for this valuable suggestion. We fully agree that a complete polarization characterization requires Stokes parameter measurements across six distinct bases. Following the suggestion from the Reviewer, we have performed additional measurements at $\pm 45^\circ$ ($H\pm V$) bases to complement the original HV and RL basis data and have updated accordingly in the revised manuscript (see Fig. R6 below for reference). This complete measurement provides a rigorous characterization of the polarization state for each harmonic order.

Fig. R6. Experimental observations and theoretical predictions of non-cascade high-harmonic random walks across six polarization bases.

Response to Reviewers' Comments:

We thank the referees for reviewing our manuscript and for their thoughtful comments and suggestions. We have revised the manuscript point by point according to their recommendations. In this response letter, we put the original *comments by the Reviewers in italics* to distinguish them from our responses in blue. Changes to the manuscript appear in red.

Reply to Reviewer #2

The authors have provided thorough and constructive responses to all comments raised by the referees. The manuscript has been significantly improved and now presents its main results in a clearer and more coherent manner. While I remain only partially convinced by the claims concerning the broader impact of the results, I find the core contribution solid and well supported.

I appreciate the novelty of the approach and the insight that the dynamics of photonic degrees of freedom in high harmonic generation can be naturally framed in terms of quantum walks. The experimental procedures appear sound and technically robust. On this basis, I recommend publication.

That said, I encourage the authors to revise the contextualization of their results. Beginning with the abstract, the comparison with the state of the art is presented from a somewhat misleading perspective. Although it is correct that cascaded systems face significant limitations for large scale integrated photonic circuits, I believe the present work should not be framed as a scalable alternative to such platforms. I am skeptical that step numbers exceeding those achievable with cascaded approaches can be realistically realized in this setting.

A clearer emphasis on the conceptual and physical innovation of the work, namely the emergence of a mechanism that can be interpreted as a quantum walk, together with the distinctive programmability enabled by pulse shaping techniques, would strengthen the manuscript and place its contributions in a more accurate context.

Reply: We thank the Reviewer for the insightful summary of our modifications and recommending the publication of our manuscript in Nature Communications. Following the kind suggestions from the Reviewer, we have rewritten the abstract in the revised manuscript as below to emphasize the natural framework of high-harmonic generation instead of the alternative to conventional cascaded platforms.

“This approach reveals the dynamics of photonic degrees of freedom in high-harmonic generation can be naturally framed as an ultrafast, high-dimensional random walk, paving the way for compact, highly stable photonic platforms tailored for solid-state information processing.” (Abstract on page 1)

Reply to Reviewer #3

All my concerns have been fully addressed, and the manuscript now meets the high standards required for publication in Nature Communications.

In terms of novelty, the authors leverage the intrinsic nonlinear optical response of a solid-state medium, using high-harmonic generation to simultaneously produce multiple harmonic orders within a single crystal. This unique approach enables parallel information processing in a compact, integrated, and inherently stable configuration, eliminating the need for cascaded components. The level of innovation demonstrated here is substantial enough to warrant publication.

Therefore, I strongly recommend that the manuscript be accepted for publication in Nature Communications.

Reply: We thank the Reviewer for kindly recommending the publication of our work in Nature Communications.